# Is there a difference in the analgesic response to intra-articular bupivacaine injection in people with knee osteoarthritis pain with or without central sensitisation? Protocol of a feasibility randomised controlled trial

Yasmine Zedan [1,2,3] Roger Knaggs,[4] Dale Cooper,[5] Thomas Kurien,[2,6] David Andrew Walsh,[2,7] Dorothee P Auer,[1,2,3] Brigitte E Scammell[2,6]

**Correspondence to**
Professor Brigitte E Scammell;
b.scammell@nottingham.ac.uk

## ABSTRACT

**Introduction** Pain is the main symptom of osteoarthritis (OA) with approximately 50% of patients reporting moderate-to-severe pain. Total knee replacement (TKR) is the ultimate treatment option to alleviate pain in knee OA. Nevertheless, TKR does not provide complete relief for all as approximately 20% of patients experience chronic postoperative pain. Painful peripheral stimuli may alter the central nociceptive pathways leading to central sensitisation that can influence treatment response in patients with OA. Currently, there is no objective protocol for detecting whether a patient will respond to a given treatment. Therefore, there is a need for a better mechanistic understanding of individual factors affecting pain relief, consequently informing personalised treatment guidelines. The purpose of this research is to examine the feasibility of conducting a full-scale mechanistic clinical trial in painful knee OA investigating the analgesic response to intra-articular bupivacaine between those with or without evidence of central sensitisation.

**Methods and analysis** The Understanding Pain mechanisms in KNEE osteoarthritis (UP-KNEE) study is a feasibility, double-blinded, placebo-controlled randomised parallel study in participants with radiographically defined knee OA and with self-reported chronic knee pain. The study involves the following assessments: (1) a suite of psychometric questionnaires; (2) quantitative sensory testing; (3) magnetic resonance imaging (MRI) scan of the knee and brain; (4) a 6-minute walk test; and (5) an intra-articular injection of bupivacaine or placebo (sodium chloride 0.9%) into the index knee. Assessments will be repeated post intra-articular injection apart from the MRI scan of the knee. Our aim is to provide proof of concept and descriptive statistics to power a future mechanistic trial.

**Ethics and dissemination** Ethical approval was obtained from the Health Research Authority (HRA) (REC: 20/EM/0287). Results will be disseminated via peer-reviewed journals and scientific conferences. The results will also be shared with lay audiences through relevant channels, such as Pain Centre Versus Arthritis website and patient advocacy groups.

---

**STRENGTHS AND LIMITATIONS OF THIS STUDY**

⇒ The study attempts to address a knowledge gap by providing mechanistic understanding of the factors contributing to the individual pain relief in knee osteoarthritis.

⇒ The feasibility design will support a future main trial.

⇒ Patients were involved in the development of the study protocol. They helped to identify the most relevant outcome measures, which were reordered to prioritise the patients' preferences.

⇒ Owing to the nature of the feasibility study, no formal hypothesis testing will be performed in this trial. It will provide descriptive statistics which will inform the design and sample size requirements of a future definitive mechanistic trial.

---

**Trial registration number** NCT05561010.

## INTRODUCTION
### Rationale

Knee osteoarthritis (OA) is a leading cause of pain, functional disability and is a substantial economic burden on society and healthcare systems.[1–3] Treatment guidelines for chronic knee pain in OA are mainly focused on pain management with a combination of pharmacological and non-pharmacological treatment approaches, but no disease-modifying treatment currently exists.[4] Total knee replacement (TKR) is the ultimate treatment to alleviate pain, nevertheless chronic pain is still reported in about 20% of patients after the surgery.[5] Currently, there is no objective protocol to detect how a patient will respond to a given OA treatment. Moreover, there is also a poor correlation between the severity

of joint damage and severity of pain,[6] which points to the contribution of central pain mechanisms.

Painful peripheral stimuli may alter the central nociceptive pathways leading to central sensitisation that can influence treatment response in patients with OA.[7] Sensitisation, which plays a key role in augmenting OA pain, is defined as a modified perception of pain due to increased impulses from the peripheral nervous system (peripheral sensitisation) along with augmentation of pain signals in the central nervous system (central sensitisation).[8] Central sensitisation was estimated to be present in about 30% of patients with OA[9] and was found to be a predictor for developing chronic postoperative pain after TKR.[10 11]

Validated proxies, including questionnaires, neuroimaging and quantitative sensory testing (QST), have been used to examine pain sensitisation. However, a gold standard to diagnose pain sensitisation in a clinical setting is lacking, and individualised mechanism-based treatment in knee OA remains a pressing need.

### Evidence gap

Given the necessity for better understanding of the mechanisms of pain relief in knee OA, the purpose of this research is to examine the feasibility of conducting a full-scale mechanistic clinical trial investigating the analgesic response in knee OA between those with or without evidence of central sensitisation. The use of intra-articular bupivacaine, as an experimental medicine approach, has been shown to significantly reduce knee pain 1 hour after injection in previous studies[12 13]; as it temporarily blocks peripheral pain stimuli, thereby potentially unmasking central pain components. Given that peripheral input commonly triggers central sensitisation, the theory that local anaesthesia will assist in differentiating patients with centrally mediated pain mechanisms is also supported by the notion that regional anaesthesia or local anaesthesia can reduce the risk of persistent postoperative pain 6 months after surgery according to a previous systematic review and meta-analysis.[14] Additionally, the use of comprehensive pain phenotyping tools including MRI of the knee and brain will enable a better mechanistic understanding of individual factors affecting pain relief, consequently informing future personalised treatment guidelines.

### Objectives

► The primary objective of the UP-KNEE study is to evaluate the feasibility of a main definitive trial by:
collecting data to inform sample size calculation;
testing the recruitment process;
Timing of the outcome measures;
Testing the robustness of randomisation;
Testing the integrity of the research protocol; and
qualitative assessment of the acceptability of the methods of the feasibility study.

► The secondary objective of the study is to explore correlations of the analgesic response with indicators of central and peripheral pain mechanisms derived from MRI of the brain and knee, questionnaires and QST.

### Hypothesis

For the future main trial, the hypothesis is that the analgesic response to a peripherally targeted intervention aiming to reduce knee pain will reveal two groups. The non-responders group will be patients with predominantly centrally driven pain characteristics, who will show less analgesic response to the peripherally targeted intervention while the responders group will be patients with predominantly peripheral pain who will show a greater analgesic response to the peripherally targeted intervention. Both groups will respond similarly to placebo. However, this feasibility study is not intended to test the hypothesis of the main trial.

## METHODS AND ANALYSIS

### Study design

The study will be a feasibility, double-blinded, placebo-controlled randomised parallel study in participants with radiographically defined knee OA and with self-reported chronic knee pain. The study design was guided by the recommendations of the Consolidated Standards of Reporting Trials (CONSORT) 2010 statement extension to randomised pilot and feasibility trials.[15] The study protocol has been reported in accordance with the Standard Protocol Items: Recommendations for Interventional Trials (SPIRIT) guidelines (online supplemental file A).[16]

### Study setting

This is a single-centre study. The study activities will be carried out at the Sir Peter Mansfield Imaging Centre, University of Nottingham.

### Sample size

This is a feasibility study aiming to provide first proof of concept and descriptive statistics to power a future diagnostic trial. We aim to recruit 50 participants (25 to each group). This would be a large enough sample to assess the feasibility and inform a main trial based on statistical methodological papers providing recommendations about sample size requirements for feasibility and pilot studies.[17–19]

### Eligibility criteria

The inclusion criteria are; patients aged 45 years and older who have the capacity to give informed consent and have radiographically defined OA knee changes (Kellgren and Lawrence Grading System K/L>2/4) with knee pain, particularly self-reported knee pain at the most severely affected side measuring between 30 and 80 mm on a 100 mm Visual Analogue Scale (VAS) for rest, use or night pain and being able to perform the 6-minute walk test (6MWT). The VAS anchors are: 0 as 'no pain' and 100 as 'the worst pain imaginable'.

The exclusion criteria are; patients having major medical, neurological and psychiatric comorbidities that would preclude completion of the study protocol. Patients who have a diagnosis of OA in any joint other than the knee with pain VAS≥30 mm, fibromyalgia, systemic or local knee infection, severe coagulopathy or taking anticoagulant therapy, known hypersensitivity to bupivacaine, taking neuropathic pain medications for their OA-related pain such as strong opioid analgesics and antiepileptic drugs will also be excluded. Known contraindications for MRI are also exclusion criteria.

## Recruitment

The trial flow is summarised in figure 1. Eligible participants will be recruited mainly through the secondary care pathway at the Trauma and Orthopaedic department in Nottingham University Hospitals (NUH) National Health Service (NHS) Trust. The clinical team directly involved in patient care will identify participants as part of their routine clinical review. In addition, there are other potential methods of patient recruitment to the UP-KNEE study, such as Primary Care pathways and participants who have previously participated in previous research in the Pain Centre Versus Arthritis and gave consent to be contacted again regarding future studies.

Informed, written consent will be obtained prior to any research activities (online supplemental file B). The participant will be given the opportunity to choose whether they consent via phone or at the appointment. Verbal consent will be taken in accordance with NUH verbal consenting guidance. Participants can withdraw from the study at any timepoint. In the event of their withdrawal, data already collected will be used, but no further data will be acquired.

## Randomisation procedure and concealment of allocation

Eligible participants will be randomised in a 1:1 allocation ratio. An online computer service (SealedEnvelope.com) will be used for stratified randomisation to either intra-articular injection of bupivacaine or sodium chloride 0.9%. This facility generates codes that are concealed from the research team to either group. To mitigate against risk of allocation imbalance with respect to whether pain is predominantly centrally or peripherally driven, stratification will be based on the scores of the Central Aspects of Pain in the Knee (CAP-Knee) questionnaire, which will be sent to participants before the research visit. Participants will be randomised on a 1:1 allocation basis to either bupivacaine or placebo. To ensure blinding of the investigators to the participant's group assignment, an authorised research nurse will prepare the allocated treatment. The randomisation schedule will be embedded by the research nurse in a secure password-protected folder to achieve allocation concealment.

## Blinding

Investigators and participants will be blind to the treatment allocation. To reduce the risk of interpretation bias, investigators analysing the data will also be blinded to the treatment allocation, and any manual image analysis will be performed by an investigator blinded to clinical status.

Emergency unblinding will only be done if emergencies occur that can be directly linked to the study medication, where it is essential to know if the patient was actually given bupivacaine.

## Experimental procedures

The study involves a single visit in which data will be collected. As per the study protocol (V.1.6, 26 January 2023), the interventions will include an assessment to check patients' suitability to undergo intervention with a local anaesthetic (bupivacaine) or placebo (sodium chloride 0.9%) intra-articular knee injection. During the visit, all participants will be invited to undertake the following assessments: (1) complete health questionnaires; (2) undergo QST; (3) MRI scan of the knee and brain; (4) perform a 6MWT; and (5) participants will then receive an intra-articular injection of bupivacaine or placebo into the index knee. Assessments will be repeated post intra-articular injection apart from the MRI scan of the knee.

Online supplemental file C contains a detailed description of the study intervention, written in accordance with the Template for Intervention Description and Replication (TIDieR) checklist and guide.[20]

The timings of the postinjection assessments were chosen based on the impact of the timing of bupivacaine administration on its analgesic effects. As the time to peak concentration is 43.4±23.1 min based on previous research using bupivacaine for peripheral analgesia,[21] there will be a 20 min interval post injection to ensure sufficient time to demonstrate the full effect of local anaesthesia and for the postprocedural care. The postinjection procedures will then be completed between 20 and 60 min post injection.

### Psychometric questionnaires

Questionnaires will be used to rate the pain severity and to evaluate psychosocial constructs. The questionnaires set include painDETECT,[22] Pain Catastrophizing Scale,[23] CAP-Knee questionnaire,[24] Beck Depression Inventory,[25] Measure of Intermittent and Constant Osteoarthritis Pain questionnaire,[26] EuroQol 5 Dimension 5 Level (EQ-5DL) questionnaire,[27] Oxford Knee Score,[28] Pittsburgh Sleep Quality Index,[29] Fatigue Severity Scale[30] and the State-Trait Anxiety Inventory.[31]

At the end of the research visit, the participant will also be asked to fill in a custom-made questionnaire with open-ended questions intended to further investigate aspects of study feasibility and acceptability of the research methods.

### 6MWT pre and post injection of bupivacaine/placebo

The self-paced 6MWT is aimed to be the functional outcome measure as it can assess the submaximal level of functional capacity. Participants will be requested to undertake a paced walk and are also allowed to stop and

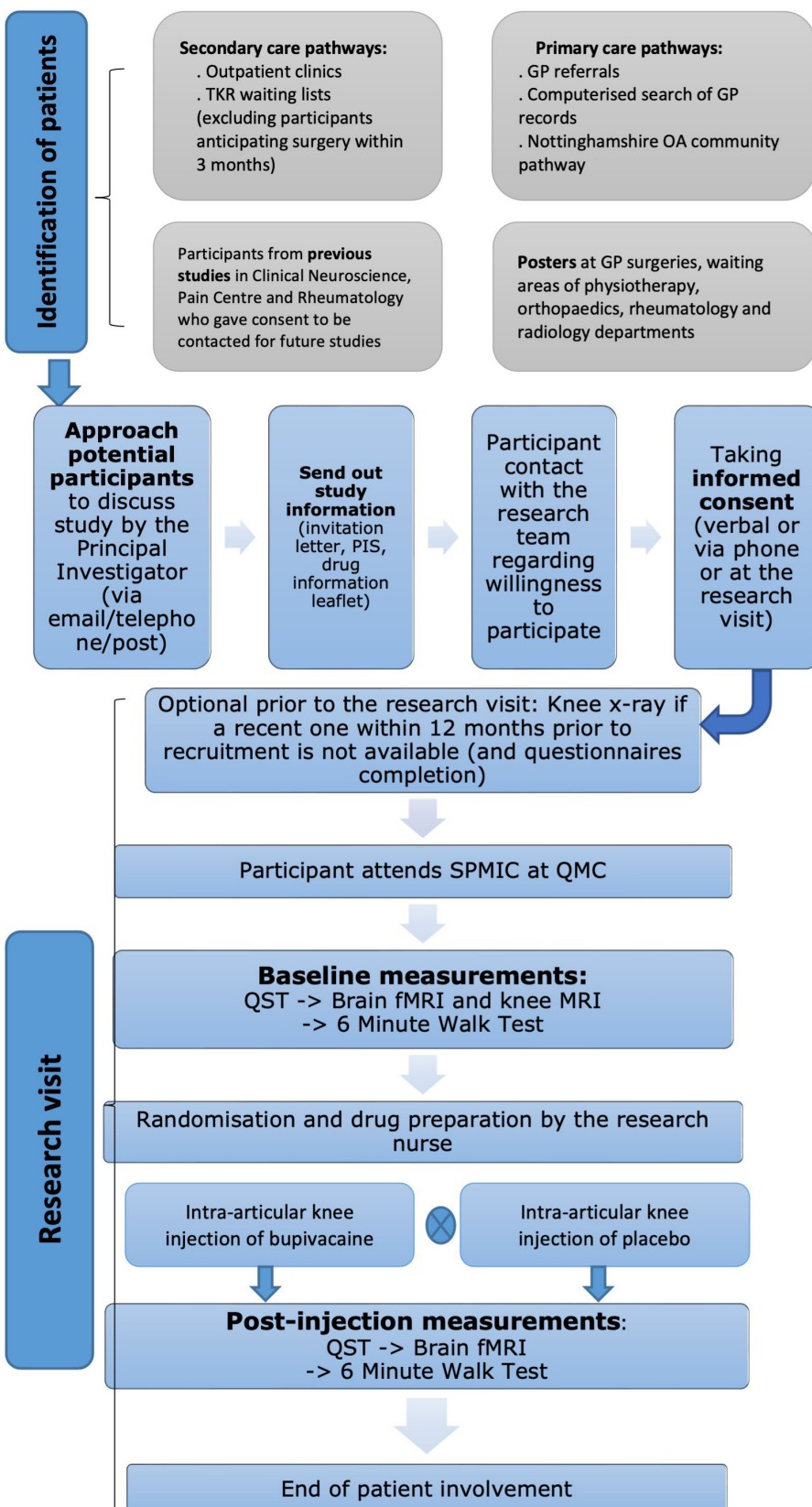

**Figure 1** Overall trial flow. fMRI, functional MRI; GP, general practice, OA, osteoarthritis; PIS, participant information sheet; QMC, Queen's Medical Centre; QST, quantitative sensory testing; SPMIC, Sir Peter Mansfield Imaging Centre; TKR, total knee replacement.

rest during performing the test. Participants will be asked to rate the level of knee pain 'pain this instant' before and immediately after the test finishes using the 0–100 VAS. The walking course of the 6MWT will be 30 m in length with a mark every 3 m.

## Quantitative sensory testing

QST is a psychophysical technique evaluating sensory response to standardised mechanical or thermal stimuli. The study will use static pain parameters; pressure pain detection threshold (PPT), and dynamic parameters; temporal summation of pain (TSP) and conditioned pain modulation (CPM). Several lines of evidence suggest that QST can be used to phenotype OA pain based on mechanisms. Enrolled patients will undergo the following QST measures: PPT, TSP and CPM, which will be performed according to a standardised methodology[32–34] (online supplemental file D).

## Intra-articular knee joint injection with bupivacaine/placebo

Bupivacaine or sodium chloride 0.9% will be administered via the intra-articular route by a member of the research team (a qualified medic) with appropriate competence in intra-articular injection and the recognition and management of the potential adverse effects. The injection will be carried out under aseptic conditions.

For intra-articular injection, the injected volume of either bupivacaine 0.25% (5 mL) or sodium chloride 0.9% (5 mL) will be the same for each participant. Bupivacaine was selected for the intra-articular injection to achieve analgesia because this was shown in previous studies to significantly reduce knee pain in participants with knee OA 1 hour after injection.[12 13]

## MRI of the knee before the injection

The participants will also undergo a short structural MRI of the knee at 3 T in order to define synovitis, bone marrow lesions and other structural pathologies in the knee joint. MRI of the knee will be evaluated using the validated semiquantitative MRI Osteoarthritis Knee Score (MOAKS).[35]

## Brain imaging

The participants will undergo a multimodal MRI scan of the brain at 3 T before and after the intra-articular injection. This study will use the functional MRI (fMRI) protocol from studies in pain imaging that can characterise spontaneous chronic pain.[36–38]

## Outcomes
### Primary outcome measures

1. The change in pain score using a 100 mm VAS during the 6MWT from baseline to 1 hour after intra-articular injection with bupivacaine or placebo.
2. The change in pain score using the VAS at rest from baseline to 1 hour after intra-articular injection with bupivacaine or placebo.

### Secondary outcome measures

1. QST: the change scores of PPT, TSP and CPM from baseline to post intra-articular injection with bupivacaine or placebo.
2. fMRI of the brain: the change in brain network activity from baseline to post intra-articular injection with bupivacaine or placebo.
3. MOAKS[35]: the level of joint damage quantified by MOAKS and measured at baseline.
4. Feasibility assessment: the number and percentage of eligible participants who are recruited and randomised to the study from the date of recruitment opening until the date of recruitment closing, as well as protocol completion rates and missing data rates.
5. Feasibility assessment: evaluation of effective randomisation of participants to the study arms using a study-specific checklist and assessment of the randomisation protocol throughout the study.
6. Feasibility assessment: a study-specific questionnaire will be administered to participants to assess the acceptability of the study at the end of the research visit.

## Data management and auditing

The collection, storage, processing and disclosure of personal information, data collection and management will comply with the requirements of the General Data Protection Regulation 2018. Handling of data will follow the policies and the procedures of the Sponsor (NUH NHS Trust) and the University of Nottingham. All personal data will be anonymised, and all further data analysis will be done without any reference to personal identifiable participant data.

The trial conduct will be monitored by the Sponsor (NUH NHS Trust).

## Statistical methods

1. Considering the feasibility nature of the study, no formal hypothesis tests will be performed to statistically compare the two study arms and descriptive statistics will be presented. Appropriate parametric or nonparametric statistics will be used according to the data characteristics.
2. Summary statistics will be used to evaluate feasibility objectives such as the feasibility to recruit, timing measurements, to describe the sample and to inform the future main trial by providing power and sample size calculation. Participant acceptability feedback will be qualitatively synthesised using thematic analysis.
3. Primary and secondary outcomes will be descriptively summarised by group as follows:
   1. Correlations between the change of pain score and QST measurements at baseline and after intra-articular injection with bupivacaine or placebo.
   2. Correlations between the change of pain score and brain network activity at baseline and after intra-articular injection with bupivacaine or placebo using predefined seeds in the pain processing regions.

3. Correlations between the change of pain score and the level of joint damage quantified by MOAKS and measured at baseline.
4. Estimates of between-group effect will be reported as estimates with 95% CIs without p values.
5. In view of the exploratory nature of the study, missing data will not be imputed. The proportion of missing data for each outcome will be described.
6. Differences in effects will be reported descriptively for primary and secondary outcomes between the following subgroups:
   1. Based on the stratification variable according to CAP-Knee Questionnaire scores with respect to whether pain is predominantly centrally or peripherally driven.
   2. The patients will be divided into responders and non-responders based on a cut-off value of ≥17 mm reduction in VAS post injection.[39]

## Safety and adverse events

The study was deemed a low-risk study by the Sponsor (NUH NHS Trust). Adverse events will be recorded and reported according to the policies of the local Research Ethics Committee and NUH NHS Trust.[40]

## Patient and public involvement

The study protocol and documents have been reshaped according to input received in two patient and public involvement (PPI) events via Pain Centre Versus Arthritis PPI advisory group. This advisory group was composed of people with OA who provided constructive feedback on the outcome measures and the study methods, which led to enhanced study design. The PPI will be maintained throughout the study and will help with dissemination of the study findings and outcomes to ensure a broader perspective.

## ETHICS AND DISSEMINATION

The study has been approved by the Nottingham Research Ethics Committee 1 (REC Reference: 20/EM/0287). The study results will be disseminated through peer-reviewed journals and communicated at scientific meetings and conferences. The results will also be presented in relevant patient websites. Authorship eligibility will be based on the recommendations from the International Committee of Medical Journal Editors (ICMJE).

## PERSPECTIVES OF THE STUDY

This feasibility randomised controlled trial will provide first proof of concept to a future main trial. The study has the scope to enhance the understanding of knee OA pain mechanisms and to pave the way for individualised treatment in knee OA.

## CURRENT TRIAL STATUS

HRA approval has been obtained. Recruitment was initiated in late 2022.

**Author affiliations**
[1]Radiological Sciences, Mental Health and Clinical Neurosciences, School of Medicine, University of Nottingham, Nottingham, UK
[2]Pain Centre Versus Arthritis, School of Medicine, University of Nottingham, Nottingham, UK
[3]Sir Peter Mansfield Imaging Centre, School of Medicine, University of Nottingham, Nottingham, UK
[4]Clinical Pharmacy Practice, School of Pharmacy, University of Nottingham, Nottingham, UK
[5]School of Allied Health Professions, Keele University, Keele, UK
[6]Academic Orthopaedics, Trauma and Sports Medicine, School of Medicine, University of Nottingham, Nottingham, UK
[7]Academic Rheumatology, School of Medicine, University of Nottingham, Nottingham, UK

**Contributors** BES conceived the study and designed the protocol with YZ, RK, DC, TK, DAW and DPA. YZ communicated with the Regulatory Authorities, initiated the study and wrote the first draft of the manuscript. All authors reviewed and approved this manuscript for submission.

**Funding** This project is supported by Versus Arthritis (Grant reference number: 20777).

**Competing interests** None declared.

**Patient and public involvement** Patients and/or the public were involved in the design, or conduct, or reporting, or dissemination plans of this research. Refer to the Methods section for further details.

**Patient consent for publication** Not applicable.

**Provenance and peer review** Not commissioned; externally peer reviewed.

**ORCID iD**
Yasmine Zedan http://orcid.org/0000-0002-6363-1374

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
