## [Reviewer comments · BMJ Open]

ARTICLE DETAILS

TITLE (PROVISIONAL)	Is there a difference in the analgesic response to intra-articular bupivacaine injection in people with knee osteoarthritis pain with or without central sensitisation?: protocol of a feasibility randomised controlled trial
AUTHORS	Zedan, Yasmine; Knaggs, Roger; Cooper, Dale; Kurien, Thomas; Walsh, David; Auer, Dorothee; Scammell, Brigitte

VERSION 1 – REVIEW

REVIEWER	Pandit, Hemant Chapel Allerton Hospital
REVIEW RETURNED	25-Feb-2023

GENERAL COMMENTS	This feasibility study is timely and well-designed. It is a mechanistic clinical trial aimed at investigating the analgesic response in knee OA in patients with and without evidence of central sensitisation. The study has a clear aim and has robust methodology. Inclusion / exclusion criteria are pragmatic and logical. The planned assessments and analyses are appropriate and employ correct tools.
--

REVIEWER	Scarvell, Jennie University of Canberra, Faculty of Health
REVIEW RETURNED	30-Mar-2023

GENERAL COMMENTS	Dear authors, This is a clearly written and clearly articulated protocol for a feasibility study. Its strengths are in its structure and clarity, and organisation. Some things the authors could consider follow. 1. Evidence gap. The background supporting the theory that local anaesthesia will assist in differentiating those people with centrally mediated pain mechanisms is clearly stated. However, the status of current research on this topic is not clear. What is the closest research to this, and what did they find? Has a review of the research been conducted?2. Consumer engagement in the development of the protocol. Patient and public involvement is noted via the advisory group meeting which has had input into the study, and is commended. Many research institutions including ethics committees and institutional research governance bodies, now stipulate that consumers should be involved in the development of research, from priority setting, to design and protocol development and interpretation of results, in a partnership. How will the partnership be maintained during the study? In a feasibility study, the input of people with lived experience is also very valuable, and the study
--

	more robust if qualitative analysis of the participants' experiences is captured. 3. The intervention. This is a protocol for a randomised clinical trial, therefore the authenticity of the intervention and placebo is critical. Thank you for the details of blinding of the clinician administering the injection, and the participant. Will the team analysing the data also be blinded to group? At what criteria would the study be unblinded? To describe intervention, the TIDier checklist is helpful. The content of the checklist appears to all be present, but it is not organised to be easy to review. The intervention and control group might be brought into a paragraph for that purpose, rather than embedding them in the procedures, among the outcomes measures. Hoffmann et al 2014 BMJ 2014;348:g1687 4. Outcomes data. Are minimal clinically important differences (MID) available for the outcome measures to be applied? Even more helpful, is smallest worthwhile effect (SWE) data available for the outcome measures? The primary outcome is pain visual analogue scale, which is well-researched. Considering MID and SWE information would strengthen the study, even though it is not the efficacy of the intervention being questioned in this study, but rather the capability of the intervention to differentiate those participants with centrally mediated pain. (Statistical analyses points 3 a,b,c) Best wishes for your study.
--	--

VERSION 1 – AUTHOR RESPONSE

Thank you for your valuable input.

In response to Reviewer 2's comments:

1. Evidence gap. The background supporting the theory that local anaesthesia will assist in differentiating those people with centrally mediated pain mechanisms is clearly stated. However, the status of current research on this topic is not clear. What is the closest research to this, and what did they find? Has a review of the research been conducted?

Response:

Given that peripheral input commonly triggers central sensitization and the well-established evidence of a neuropathic component in OA, the theory that local anaesthesia will assist in differentiating patients with centrally mediated pain mechanisms is also supported by the notion that regional or local anaesthesia can reduce the risk of persistent postoperative pain six months after surgery according to a previous systematic review and meta-analysis.[1] Regional analgesics and local anaesthetics can also provide pain relief in neuropathic pain conditions, such as occipital neuralgia and meralgia paresthetica. This provides proof-of-concept validation for the approach suggested in our study as bupivacaine would be expected to reduce pain from neuropathy within the joint and help differentiate patients with centrally mediated pain mechanisms. To the best of our knowledge, no study has investigated this approach in knee OA, hence the novelty of our study.

[1] Andreae MH, Andreae DA. Regional anaesthesia to prevent chronic pain after surgery: a Cochrane systematic review and meta-analysis. Br J Anaesth. 2013 Nov;111(5):711-20

2. Consumer engagement in the development of the protocol. Patient and public involvement is noted via the advisory group meeting which has had input into the study, and is commended. Many research institutions including ethics committees and institutional research governance bodies, now stipulate that consumers should be involved in the development of research, from priority setting, to

design and protocol development and interpretation of results, in a partnership. How will the partnership be maintained during the study? In a feasibility study, the input of people with lived experience is also very valuable, and the study more robust if qualitative analysis of the participants' experiences is captured.

Response:

Thank you for your comment. Patient and public engagement will be maintained during the study through the Pain Centre Versus Arthritis PPI advisory group which, in addition to their input to the study design, will also be involved in the dissemination of the study findings and outcomes. We have updated the manuscript to clarify this point.

To answer the second part of your question, qualitative analysis of the participants' experiences will be captured through a thematic analysis of the participant acceptability feedback as mentioned in the 'Outcomes' and the 'Statistical methods' sections.

3. The intervention. This is a protocol for a randomised clinical trial, therefore the authenticity of the intervention and placebo is critical. Thank you for the details of blinding of the clinician administering the injection, and the participant. Will the team analysing the data also be blinded to group? At what criteria would the study be unblinded?

To describe intervention, the TIDier checklist is helpful. The content of the checklist appears to all be present, but it is not organised to be easy to review. The intervention and control group might be brought into a paragraph for that purpose, rather than embedding them in the procedures, among the outcomes measures.

Hoffmann et al 2014 BMJ 2014;348:g1687

Response:

Thank you for your comment and for the useful suggestion regarding using the structured TIDier checklist.

The team analysing the data will also be blinded to the treatment allocation. This has been clarified in the 'Blinding' section.

As for unblinding, planned unblinding will be performed at the end of the study after finalizing the statistical analysis and only with the agreement of the sponsor. Emergency unblinding will only be done if emergencies occur that can be directly linked to the study medication, where it is essential to know if the patient was actually given bupivacaine.

We have attached an appendix containing a detailed description of the intervention, written in accordance with the TIDieR checklist.

4. Outcomes data. Are minimal clinically important differences (MID) available for the outcome measures to be applied? Even more helpful, is smallest worthwhile effect (SWE) data available for the outcome measures? The primary outcome is pain visual analogue scale, which is well-researched. Considering MID and SWE information would strengthen the study, even though it is not the efficacy of the intervention being questioned in this study, but rather the capability of the intervention to differentiate those participants with centrally mediated pain. (Statistical analyses points 3 a,b,c)

Response:

Previous research has estimated an MCID in PPT (one of the QST parameters which is a secondary outcome measure in our study) of the lumbar paraspinal muscles to be around 114 kPa in healthy volunteers which is a different context and may not be pertinent to knee OA. However, to the best of our knowledge, an MCID/SWE for other secondary outcome measures in our study has not been identified in OA patients.

In response to the Editor's comments:

- Along with your revised manuscript, please include a copy of the SPIRIT checklist indicating the page/line numbers of your manuscript where the relevant information can be found (<http://www.spirit-statement.org/>)

Response:

A copy of the SPIRIT checklist has been attached as supplementary material.

- Along with your revised manuscript, please provide an example of the participant consent form as a 'Supplemental Material' file, as per item #32 of the SPIRIT checklist.

Response:

The participant consent form has been attached as supplementary material.

- Please ensure that the information provided in your protocol article is consistent with that included in the trial registry. For example, the secondary outcomes. Please update the manuscript and/or trial registry accordingly.

Response:

The outcomes section of the trial registry is being updated to be consistent with the format of the protocol article.

VERSION 2 – REVIEW

REVIEWER	Scarvell, Jennie University of Canberra, Faculty of Health
REVIEW RETURNED	26-Jun-2023
GENERAL COMMENTS	Best wishes for the successful completion of your project.